# A Mini Review of Novel Topoisomerase II Inhibitors as Future Anticancer Agents

**DOI:** 10.3390/ijms24032532

**Published:** 2023-01-28

**Authors:** Cosmas O. Okoro, Toluwase Hezekiah Fatoki

**Affiliations:** Department of Chemistry, Tennessee State University, Boswell Science Complex, Nashville, TN 37209-1561, USA

**Keywords:** mini, topoisomerase II, catalytic inhibitors, poisons, DNA, salicylate, cardiotoxicity, MD simulation

## Abstract

Several reviews of inhibitors of topoisomerase II have been published, covering research before 2018. Therefore, this review is focused primarily on more recent publications with relevant points from the earlier literature. Topoisomerase II is an established target for anticancer drugs, which are further subdivided into poisons and catalytic inhibitors. While most of the topoisomerase II-based drugs in clinical use are mostly topoisomerase II poisons, their mechanism of action has posed severe concern due to DNA damaging potential, including the development of multi-drug resistance. As a result, we are beginning to see a gradual paradigm shift towards non-DNA damaging agents, such as the lesser studied topoisomerase II catalytic inhibitors. In addition, this review describes some novel selective catalytic topoisomerase II inhibitors. The ultimate goal is to bring researchers up to speed by curating and delineating new scaffolds as the leads for the optimization and development of new potent, safe, and selective agents for the treatment of cancer.

## 1. Introduction

Cancer is a word that creates deep-seated fear because we immediately associate it with grave illness and high mortality rate. Almost all of us know someone whose life has been blighted by cancer diagnosis and who has suffered the prolonged pain of the illness. Cancer patients are forced to tolerate a tough treatment regime with all the accompanying side effects. Few are fortunate to escape the distress of cancer over their lifetime since the frightening statistics suggest that the vast majority of us will either experience it firsthand or have a loved one afflicted by it. As a result of the above facts, and in order to improve the survival and quality of life of cancer patients, medicinal chemists are actively searching for novel effective, safe, and selective anticancer drugs. This review provides an overview of the chemical structures and bioactivities of recent agents that target the nuclear enzyme, topoisomerase. Essential nucleic acid functions, such as DNA replication and recombination, generate knots and tangles within the double helix. DNA knots are known to impair the ability to separate the two strands. In addition, intermolecular DNA tangles prevent the segregation of chromosomes during mitosis. It is now recognized that topoisomerases regulate the topological structure of DNA. If the above topological obstacles are not removed, they become lethal to cells. Topoisomerases fall into two major classes, topoisomerase I and topoisomerase II. The classification is based upon the linking number changed by the enzyme: type 1 changes the linking number by one while type II changes the linking number by two. Topoisomerase I breaks one strand of the double helix and its functions includes regulation of the levels of DNA supercoiling. On the other hand, topoisomerase II breaks both strands of the double helix and regulates the super-helical density and removal of knots and tangles in the duplex DNA. To maintain genomic integrity during the required DNA cleavage event, all topoisomerases form covalent bond between the active-site tyrosyl residues and the DNA termini generated during the reaction. This review focuses on topoisomerase II (topo II). There are two isoforms of human topo II, topo IIα and topo IIβ. Both isoforms are encoded by separate genes but appear to share approximately 70% of amino acid sequence identity. However, both isoforms have distinct patterns of expression. Topo IIα has the following characteristics: (1) it is essential for the survival of proliferating cells; (2) its protein levels rise dramatically during periods of cell growth; (3) it is regulated over the cell cycle with the concentration peaking in the G2/M phase; (4) it helps alleviate the torsional stress that accumulates ahead of the replication forks and transcription complexes; and (5) it is required for proper chromosome condensation, cohesion, and segregation. Topo IIβ has the following features: (1) it is dispensable at the cellular level; (2) its concentration is independent of cell cycle; (3) high levels of Topo IIβ are found in most cell types regardless of proliferation status; and (4) it dissociates from chromosome during mitosis. The physiological functions of topo IIβ are yet to be fully understood.

Topoisomerase (Topo) is an established target for anticancer drugs and is known to be responsible for regulating the topological constraints in DNA, such as under-winding, over-winding, knotting, and tangling. In particular, under-winding (negatively supercoiled) is important because the two strands of DNA must be separated in order for replication to start. Thus, one of the physiological roles of topoisomerase is to relax both positive and negative supercoiled DNA [1]. Topoisomerase falls into two major classes in human eukaryotes, namely type I and type II. Type I topoisomerase has the subfamilies type IA (Topo IIIα and Topo IIIβ) and type IB (Topo I and mtTopo 1), while type II topoisomerase has the subfamily type IIA (Topo IIα and Topo IIβ) [2]. Human Topo I is a monomeric protein that does not need cofactors for biological activity. It regulates the topological state of DNA by breaking a single strand and re-ligating it after the strand passage reaction. Human Topo II is a homodimer that requires Mg^2+^ and ATP for its catalytic activity. Topo II breaks double strand and passes an intact DNA through the break. 

Topo II inhibitors are classically divided into catalytic inhibitors and topo II poisons, according to their mechanism of action. Topo II catalytic inhibitors destroy cancer cells through the inhibitions of Topo II enzymatic activities, thus preventing the formation of Topo II–DNA complex without increasing DNA cleavage, via the mechanisms of action that include interfering with DNA binding, inhibiting cleavage of the DNA molecule, ATP hydrolysis, and binding to the ATP binding site. Topo II poisons destroy cancer cells by increasing the amount of covalent Topo II–DNA complexes and preventing the religation of the cleaved DNA strands, thus forming unwanted double strand breaks that are toxic to the cells, and, subsequently, leading to apoptosis. Based on the type of DNA binding strategies, Topo II poisons could be DNA-intercalating agents, which have a weak interaction with DNA and function by insnaring Topo II–DNA complexes, or non-DNA-intercalating agents, which can reversibly incorporate themselves into the DNA base pairs and hinder the activity of enzymes responsible for DNA replication and transcription processes. Topo II poisons include drugs such as etoposide **1**, doxorubicin **2**, and m-amsacrine **3** [3,4,5], as shown in Figure 1. Most of the first-line agents for treating cancer are Topo II poisons, such as etoposide (non-intercalator), doxorubicin, and m-amsacrine (intercalator) [4,5,6,7,8]. However, a study by Ketron et al. [9] indicated that the activity and specificity of m-amsa lies in the head group 4′-amino-methanesulfon-m-anisidide. The DNA intercalation of m-amsa is used primarily to enhance the affinity of the drug for Topo II–DNA cleavage complex. However, due to side effects, such as risk of cardiotoxicity and secondary malignancies, that are often encountered during the use of DNA poisonous drugs, research is now shifting towards the discovery of Topo II catalytic inhibitors, which have good pharmacokinetics profiles. 

Topo II is further divided into Topo IIαand Topo IIβ. Topo IIα is highly expressed in proliferating cells, while Topo IIβ is dispensable during proliferation. Although the two isoforms are structurally similar, the precise role of the β-isoform is unclear. There is speculation that the B-isoform is responsible for the occurrence of acute MLL in patients treated with a topoisomerase inhibitor. Topoisomerase II has four domains: (i) an N gate, (ii) a DNA gate, (iii) a C gate, and (iv) a C-terminal domain that is responsible for DNA recognition [10,11]. 

Topoisomerase II catalytic cycle consists of six sequential steps as follows: (i) recognition and binding of the enzyme to DNA helix 1; (ii) trapping of DNA helix 2 through the ATPase domain dimerization; (iii) double-strand break that results in covalent bond formation between the enzyme and the 5′ phosphodiester of DNA; (iv) another segment of DNA is passed through the break facilitated by ATP hydrolysis; (v) religation of the DNA strand break mediated by the release of ADP; and (vi) the enzyme and DNA are each restored and ready to start another catalytic cycle. It is worthy of note that DNA cleavage by either Topo I or Topo II is transient and rate determining, while the religation process is fast and well tolerated by cells. The ability of small organic molecules to modulate topoisomerase activity is an effective method for identifying new cancer therapeutics. On the following pages, we delineate recent inhibitors that are still in various stages of development.

A study that used virtual high-throughput screening (VHTS) of the ZINC database showed that four zinc compounds, **4**–**7**, could be potent inhibitors of TopoIIα based on their better docking score than the standard drug etoposide, as well as their suitable predicted ADME/Tox properties [12]. Similarly, an in silico study conducted in Nigeria by Adeniran et al. [13], which used VHTS, three-dimensional quantitative structure activity and relationship (3D-QSAR), and molecular docking approaches, reported the potential of 20-betaecdysone **8** and andropanoside **9** as better inhibitors of topoisomerase IIα (TopoIIα) than the standard drug, etoposide. This study needs further investigation as these compounds are phytochemicals and could be structurally optimized to deliver efficient anticancer activity. Additionally, in Slovenia, Skok et al. [14] used in silico screening of bacterial topoisomerase inhibitors with in vitro assay to identify ATP-competitive inhibitors of human DNA TopoIIα, and they reported *N-(4-Carbamoyl-2-isopropoxyphenyl)-3,4-dichloro-5-methyl-1H-pyrrole-2-carboxamide* as a potential active inhibitor of TopoIIα. Further investigation of these computationally screened compounds is necessary to validate their biological activity as anticancer agents.

Ellipticine, an alkaloid from *Ochrosia elliptica* labil, has been previously indicated for the treatment of metastatic breast cancer, and several carbazole derivatives based on Ellipticine have shown inhibitory activities against TopoI and TopoII [15,16,17]. In Italy, Saturnino et al. [18] reported *2-(4-((3-Chloro-9H-carbazol-9-yl)pentyl)piperazin-1-yl)-N,N,N-trimethylethanammonium iodide*
**10** as a good inhibitor of TopoII and that it showed antiproliferative activity on breast cancer cells, causing apoptosis by activating the caspase pathway. Additionally, *7-((2-(dimethylamino)ethyl)amino)indolo[2,1-b]quinazoline-6,12-dione* **26**, which was derived from trypthantrin, a natural alkaloidal compound containing a basic indoloquinazoline moiety, has been reported for its inhibitory activity against TopoII, and has shown properties such as high water solubility and antiproliferative activity on acute leukemia, colon, and breast cancer cell lines [19]. Garcinol, a polyisoprenylated benzophenone isolated from the *Garcinia* genus, has been reported to inhibit eukaryotic TopoI and TopoII at concentrations comparable to that of the standard drug etoposide [20]. Mai et al. [21] in China investigated *9-bromo-2,3-diethylbenzo[de]chromene-7,8-dione* (MSN54) (**16**), a derivative of Mansonone F (MsF) [22], and they found it to be a non-intercalative catalytic inhibitor of TopoIIα. MsF (**17**) is a phytoalexin obtained from *Helicteres Angustifolia* L. with a rare sesquiterpene o-quinone structure and possesses anti-tumor activity, and its derivatives have shown strong inhibitory activity on TopoII [23]. 

A study in Italy that made use of biochemical assays along with molecular docking and molecular dynamic methods reported that the derivatives of garcinol acted as catalytic inhibitors of TopoII via a mixed inhibition of ATP hydrolysis, in which guttiferone M showed the most significant effects against TopoIIα and TopoIIβ, whereas oxyguttiferones K and M showed activity against TopoIIβ that was slightly better than that of the parent garcinol compound [24]. A collaborative work by Zidar et al. [25], in Italy and Slovenia, reported that *N-phenoxylypropyl-3-methyl-2-phenyl-1H-indoles*
**20** and **21** had potent antiproliferative and anti-topoisomerase II activities against three selected human tumor cell lines, including cervix adenocarcinoma (HeLa), ovarian carcinoma (A2780), and biphasic mesothelioma (MSTO-211H), and they were capable of inducing the apoptosis pathway. These compounds have structural similarities with some naturally occurring flavonoids that inhibit topoisomerase II (TopoII), such as quercetin and luteolin. Thus, they are scientifically acceptable and could have broader application in anticancer drug development.

Zhou et al. [26], in China, studied novel perimidine *o*-quinone derivatives and found that *2-(4-Chlorophenyl)-1-methyl-1H-perimidine-5,6-dione* showed the best antiproliferative activity (IC50 ≤ 1 μM) against four cancer cell lines (HL-60, Huh7, Hct116, and Hela) by inducing apoptosis in a dose-dependent manner, while exhibiting potent topoisomerase IIα (TopoIIα) inhibitory activity (IC_50_ = 7.54 μM). They provided evidence that compound 21 did not intercalate into DNA and suggested that it might act as an ATP competitive inhibitor by blocking the ATP-binding site of the enzyme; this was also tested by molecular docking of compound 21 in the ATP-binding domain of human TopoIIα (PDB code: 1ZXM). Overall, this study is robust and may provide advanced opportunities for the design and development of new chemotherapeutic agents.

Sakr et al. [27], in Egypt, reported that *N-Cyclohexyl-2-(3-methyl-[1,2,4]-triazolo[3,4-a]phthalazin-6-yl)-hydrazine-1-carboxamide* **11**, a derivative of triazolophthalazine, showed slightly high cytotoxic activity than doxorubicin when tested against human cancer cell line (HepG2, MCF-7, and HCT-116 cells), induced apoptosis in HepG2 cells and G2/M phase cell cycle arrest, and showed Topo II inhibitory activity. However, they reported that compound 22 showed TopoII poisoning effects at 2.5 μM and Topo II catalytic inhibitory effects at 5 and 10 μM; these results indicate that this compound could serve as a two-edge chemotherapeutic agent, thus requiring further validation. A previous study reported that a compound that contained the 1,4-diaminobenzo[g]phthalazine nucleus had a promising binding affinity against DNA by intercalation [28]. In addition, Arencibia et al. [29] reported *6-Hydroxy-4-oxo-1,3-diphenyl-2-thioxo-N-(3-(trifluoromethoxy)phenyl)-1,2,3,4-tetrahydropyrimidine-5-carboxamide* **24** as the most promising drug-like candidate that acted as a TopoII poison and exhibited good solubility, metabolic (microsomal) stability, and promising cytotoxicity in three cancer cell lines (DU145, HeLa, and A549).

Additionally, four compounds from a new series of [1,2,4]triazolo[4,3-*a*]quinoxaline and bis([1,2,4]triazolo)[4,3-*a*:3’,4’-*c*]quinoxaline derivatives showed cytotoxic activities against three tumor cell lines (HePG-2, Hep-2, and Caco-2), of which *2-(Bis[1,2,4]triazolo[4,3-a:3’,4’-c]quinoxalin-3-ylsulfanyl)-N-(4-fluorophenyl) acetamide* was the most effective inhibitor against TopoII, intercalated DNA, caused cell cycle arrest at the G2/M phase, and induced apoptosis in Caco-2 cells [30]. Moreover, El-Adl et al. [31], in Egypt, worked on twenty four novel [1,2,4]triazolo[4,3-*a*]quinoxaline derivatives, and they reported that *2-{4-([1,2,4]Triazolo[4,3-a]quinoxalin-4-ylamino)benzoyl}-N-cyclohexylhydrazine-1-carboxamide*
**12**, *2-{4-([1,2,4]Triazolo[4,3-a]quinoxalin-4-ylamino)benzoyl}-N-cyclohexylhydrazine-1-carbothioamide*, and *4-(Diethylamino)-[1,2,4]triazolo[4,3-a]quinoxaline-1-thiol*
**11b** were the most potent derivatives against the tested HepG2, HCT116, and MCF-7 cancer cell lines, and that these three compounds also displayed very-good-to-moderate DNA-binding affinities and exhibited very good inhibitory activities against TopoII enzyme. However, no in vitro differentiation was made between TopoIIα and TopoIIβ, although molecular docking was carried out on DNA-Topo II receptor (PDB code: 4G0U). Previous studies had reported that several quinoxaline compounds, such as *1-(2-Bromoethyl)-1,4-dihydroquinoxaline-2,3-dione, 4-Amino-N’-(3-chloroquinoxalin-2-yl) benzohydrazide, N’-(3-chloroquinoxalin-2-yl)-isonicotinohydrazide,* and *3-mercaptoquinoxalin-2-yl carbamimidothioate*, from the series of quinoxaline derivatives were DNA intercalators, effective inhibitors of TopoII, and showed antiproliferative activities against HePG-2, MCF-7, and HCT-116 cell lines [32,33,34,35].

Bruno et al. [36], in USA, explored the CRISPR datasets (19Q3 DepMap Public data) together with biochemical and cell biological assays and showed that CX-5461 **25,** which is structurally similar to ciprofloxacin and voreloxin, exerts its primary cytotoxic activity through topoisomerase II poisoning. This study was holistic in its approach, and it could be applied to other investigational small molecules. The authors attempted to rethink the verdict by Lin et al. [37], which stated that off-target toxicity is a common mechanism of action of cancer drugs undergoing clinical trials. They further suggested that the mechanism of cell death induced by CX-5461 is critical for rational clinical development in patients with relapse/refractory hematopoietic tumors based on its previous clinical indication as an inhibitor of RNA polymerase I [38,39]. Recent study has suggested that CX-5461 could stabilize G-quadruplex DNA and cause DNA damage [40], and that BMH-21, which is an inhibitor of RNA polymerase I, suppresses the nucleolar translocation of both TopoIIα and TopoIIβ in ATP-depleted cells [41]. Similarly, in United Kingdom, Cowell et al. [42] explored PR-619 [*2,6-Diaminopyridine-3,5-bis(thiocyanate)*] **31**, a broad-spectrum deubiquitinating enzyme (DUB) inhibitor [43], and they found it to be a potent TopoII poison, inducing both TopoIIα and TopoIIβ covalent DNA complexes with an efficiency equal to that of etoposide.

On prostate cancer (PCa)-targeted TopoII inhibition, Jeon et al. [44] investigated the mechanism by which AK-I-190 [*2-(3-trifluorophenyl)-4-(3-hydroxyphenyl)-5H-indeno[1,2-b]pyridin-6-ol*] inhibited TopoII by using various types of biological and spectroscopic evaluations, and they found that its inhibitory activity was through intercalating into DNA without stabilizing the DNA–enzyme cleavage complex, which resulted in significantly less DNA toxicity than etoposide and inhibited the growth of AR-negative PCa cells. This work served as a therapeutic strategy against castration-resistant prostate cancer (CRPC), resulting from androgen independence in cellular growth [45]. The lead compound, T60 (PubChem CID: 36589274) **13**, was reported as a potent inhibitor of both TopoIIα and TopoIIβ enzymatic activities, as well as having dual inhibitory activity on the androgen receptor (AR) (an oncogenic transcriptional factor that requires TopoIIβ) and AR-positive PCa cell growth [46]. Matias-Barrios et al. [47] in Canada investigated the derivatives of T60 in order to improve its pharmacokinetic properties and further enhance its efficacy to inhibit TopoII proteins, and they found that T638, an amino derivative on the central benzene ring of T60, retained TopoII inhibitory activities and showed improved solubility and better metabolic stability, with the possibility of high dosage administration due to low cytotoxicity [47]. In a similar manner, heteronemin (a marine sponge *Hyrtios* sp. Sesterterpene) promoted apoptosis and autophagy through the inhibition of TopoIIα and HSP90 as well as through protein tyrosine phosphatase (PTP) activation in PCa cells [48], and inhibited TNFα-induced NF-κB activation through proteasome and induced apoptotic cell death [49]. 

Ortega et al. [50] in Italy studied a novel class of 6-amino-tetrahydroquinazoline derivatives, and they pinpointed *N^4^-[4-(Dimethylamino)phenyl]-2-(4-pyridyl)-5,6,7,8-tetrahydroquinazoline-4,6-diamine* **15** as the main lead compound for the inhibition of DNA relaxation, which possessed excellent metabolic stability and solubility than etoposide; this compound showed about 100-fold selectivity for TopoIIα over TopoIIβ, with a broad antiproliferative activity against cultured human cancer cells, a satisfactory in vivo pharmacokinetic profile, and penetrability of the blood−brain barrier. These excellent properties indicated this compound 36 as a highly promising lead for the development of novel and potentially safer TopoII-targeted anticancer drugs. Additionally, in China, Chen et al. [51] evaluated the derivatives of acridine hydroxamic acid, and they found that *7-(4-(4-(Acridin-9-ylamino)-phenyl)-1H-1,2,3-triazol-1-yl)-N-hydroxyheptanamide*
**27** showed the best inhibitory activities against TopoII and histone deacetylase (HDAC), and it could intercalate into DNA and induce U937 apoptosis. A combination of Topo and HDAC inhibitors has been found to show synergistic anticancer effects with enhanced cytotoxicity [52,53]. These dual inhibitory compounds are promising drug candidates that could serve as a double-edge sword to effectively inhibit tumor growth and progression. 

In USA, Oyedele et al. [54] worked on a novel series of acridone derivatives, of which five derivatives, including 7-chloro-3-phenyl-3,4-dihydroacridin-1(2H)-one], 7-bromo-3-phenyl-3,4-dihydroacridin-1(2H)-one, 7-methoxy-3-(trifluoromethyl)-3,4-dihydroacridin-1(2H)-one, 7-methoxy-3-phenyl-3,4-dihydroacridin-1(2H)-one, and 5,7-dibromo-3-phenyl-3,4-dihydroacridin-1(2H)-one, showed excellent in vitro antiproliferative activities against 60 human cancer cell lines. Overall, 5,7-dibromo-3-phenyl-3,4-dihydroacridin-1 (2H)-one was found to be the most active and sensitive agent in all the nine cancer panels in the order of prostate > leukemia > non-small cell lung cancer > colon cancer > CNS cancer > melanoma > renal cancer > ovarian cancer > breast cancer, and the authors suggested possible inhibition of TopoIIα based on molecular binding interaction with the active site of the ATPase domain [54]. A limitation of this work was that no standard drug was used to compared against the activity of the synthesized compounds on the nine cancer panels, and an actual assay for TopoIIα was not conducted. Moreover, in China, Li et al. [55] reported that a newly developed acridone derivative, 1-((3-(dimethylamino)propyl)amino)-7-hydroxy-4-nitroacridin-9(10H)-one, could inhibit TopoIIα, intercalated with DNA, and showed significant and long-term antiproliferative activity at relatively high concentrations. Previous studies have identified several acridone derivatives as TopoII inhibitors and DNA intercalators with cell cycle arrest and apoptosis [56,57]. Furthermore, in Egypt, Nemr et al. [58] and Nemr and AboulMagd [59] worked on a novel series of thiazolopyrimidines and fused thiazolopyrimidines, and screened for anticancer activity against 60 human cancer cell lines. They found Ethyl 4-(4-bromophenyl)-2-imino-9-(3,4,5-trimethoxyphenyl)-7-phenyl-1,2-dihydro-9H-pyrimido[4’,5’:4,5]thiazolo[3,2-a]pyrimidine-8-carboxylate and Ethyl 3-(4-chlorophenyl)-5-(4-chlorophenyl)-7-phenyl-5H-thiazolo-[3,2-a] pyrimidine-6-carboxylate hydrobromide to be potent inhibitors against a renal cell line (A-498) and induce cell cycle arrest at the G2/M phase, leading to cell proliferation inhibition and apoptosis, and their fused derivative both showed potent TopoII inhibitory activity, with IC_50_ slightly higher than that of the standard drug, doxorubicin.

Moreover, in Egypt, *2-(1-Ethyl-7-methyl-4-oxo-1,4-dihydro-1,8-naphthyridine-3-carbonyl)-N-(m-tolyl)-hydrazinecarbothioamide,* a derivative of nalidixic acid (**14**), has been shown to be a potent inhibitor of TopoIIα and TopoIIβ and induce cell cycle arrest at the G2-M phase, leading to inhibition of cell proliferation and apoptosis [60]. According to Jiang et al. [61], four compounds from a series of carbazole-rhodanine conjugates were found to possess topoisomerase II inhibitory activity, with potency at 20 µM. However, this study did not use any human cell lines but used plasmid (pBR322 DNA), did not extensively explore biological activities, and failed to differentiate between TopoIIα and TopoIIβ. Shrestha et al. [62] in Korea investigated a series of new benzofuro[3,2-b]pyridin-7-ols derivatives, and their results showed a chemical structure named compound 11, that has *meta*-OH positions in the 2,4-diphenol moieties of benzofuro[3,2-b]pyridin-7-ol ring, as having the most selective and potent TopoII inhibition, with the sturdiest antiproliferative activity in HeLa cell line. Although this work could differentiate between Topo I and TopoII activities, it failed to classify the inhibition of TopoII as whether coming from IIα or IIβ. A collaborative work by Oviatt et al. [63] in Italy and USA reported that etoposide derivatives, in which the C4 sugar moiety was replaced with a variety of polyamine tails, induced higher levels of DNA cleavage with human topoisomerases IIα and IIβ than did the parent drug. Although some of the hybrid compounds showed better cleavage on TopoIIα than etoposide, the interaction of all these derivatives on TopoIIβ showed a greater fold cleavage than etoposide, implicated Gln778, and limited their further clinical usefulness.

In China, Song et al. [64] worked on a novel series of pyrazoline **22**, **23** derivatives, and they reported 8-chloro-3-(1H-indol-3-yl)-2-phenyl-2,3,3a,4-tetrahydrothiochromeno[4,3-c] pyrazole and 6,8-dichloro-3-(1H-indol-3-yl)-2-phenyl-2,3,3a,4-tetrahydrothiochromeno-[4,3-c]pyrazole as having antiproliferative activity in four human cancer cell lines (MGC-803, Hela, MCF-7, and Bel-7404) and a low cytotoxicity in the normal cell line L929 in vitro. Both were non-intercalative Topo II catalytic inhibitors and were able to induce G2/M cell cycle arrest and apoptosis in MGC-803 cells. Moreover, the derivatives of 4,5′-bithiazoles were reported to be acting as catalytic inhibitors of TopoIIα via the competitive inhibition of ATP hydrolysis, and they were able to reduce cell proliferation and stop the cell cycle mainly in the G1 phase [65]. Li et al. [66] in China reported that N-(3-(4-Methylpiperazin-1-yl)propyl)-50-methyl-10H-ursa-2,12-dieno[3,2-b]indol-28-carboxamide, a new indole derivative of ursolic acid, exhibited the most effective activity against two human hepatocarcinoma cell lines (SMMC-7721 and HepG2) and normal hepatocyte cell line (LO2) via a MTT assay. The results showed that the compound significantly inhibited TopoII activity, elevated intracellular ROS levels, decreased mitochondrial membrane potential, and caused apoptosis of SMMC-7721 cells.

Moreover, Legina et al. [67] in Austria used biological assays and molecular dynamic simulations to show that thiomaltol-containing ruthenium (Ru^II^)-, osmium (Os^II^)-, rhodium (Rh^III^)-, and iridium (Ir^III^)-based organometallic complexes bearing 1-methylimidazole or chloride as the leaving group possessed cytotoxic and DNA-damaging activity in human mammary carcinoma cell lines. A study on the anticancer properties of novel Ru^II^, Os^II^, Rh^III^, and Ir^III^ thiomaltol complexes showed that they acted as inhibitors of TopoII catalytic activity and had a significantly higher enzyme inhibitory capacity than the free ligand [68]. In 2014, Bau, Kang and their group [69] reported that salicylate **29** showed selectivity for topo IIα-isoform in DNA cleavage assay, thus acting as a catalytic inhibitor. However, further studies are needed to confirm the basis for its isoform selectivity. In their studies, they reported that salicylate did not intercalate DNA and did not prevent the enzyme from interacting with DNA. Furthermore, salicylate did not stabilize the cleavable “complex”.

Ziga et al. [14] reported **30** as a novel ATP-competitive inhibitor of hDNA topo IIα containing pyrrolamide pharmacophore. The compound showed high kinetic ATPase activity (IC_50_ 0.43 µM). In another study, Kamila et al. [70] reported that **32**, a quinolone derivative, is being used in phase I and II clinical trials in combination with azacitidine and infusional cytabarin. Recently, Sisodiya et al. [71] reported the synthesis of benzo-fused carbazolequinone derivatives that contain both indole and quinone moieties that are found in numerous drugs, including natural products. Compound **28** displayed significant apoptotic antiproliferation in cancer cells with cell cycle arrest at the S phase. It also inhibited topoIIα with more efficiency compared to etoposide. The structures of topoisomerase II inhibitors (**4**–**17** and **18**–**32**) are shown in Figure 2 and Figure 3.

## 2. Conclusions

In this review we see a significant number of reports of small molecule inhibitors of topoisomerase II. Topoisomerase II poisons continue to dominate the literature despite reports of cardiotoxicity and multi-drug resistance, including secondary malignancy. In the past three years, we have also seen a gradual increase in the number of catalytic inhibitors, which appear more attractive from a safety standpoint. However, no catalytic inhibitor has received FDA approval. The large number of reports of topoisomerase II inhibitors in the recent literature reflect the high level of interest in topoisomerase II inhibitors as therapeutic targets. Several reports have confirmed the structural similarity between topo IIα and topo IIβ. The two isoforms are similar except in the C-terminus. The above calls for structure-based drug design beyond molecular docking. Docking simulations are prone to inaccuracy because the scoring functions used make estimates of the binding energy. In addition, docking often excludes hydrogens and solvents. On the other hand, molecular dynamic simulation of the drug–protein complex gives a more accurate binding energy since it considers protonation and solvation. Thus, docking and MD simulation would undoubtedly reveal the structural differences that exist at the C-terminus. In addition, structural modification of natural products and hit molecules will continue to be an integral part of the drug discovery of novel topoisomerase II inhibitors.

## Figures and Tables

**Figure 1 ijms-24-02532-f001:**
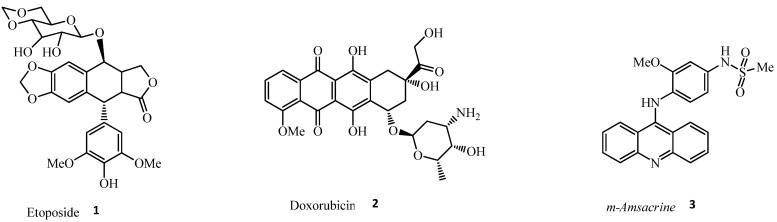
Structure of topoisomerase Ⅱ poisons.

**Figure 2 ijms-24-02532-f002:**
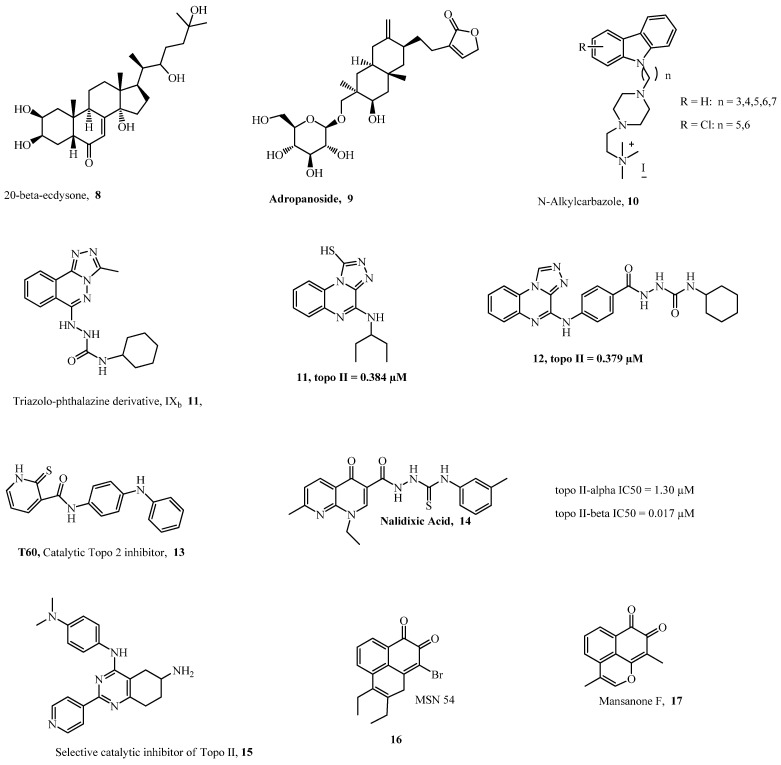
New Topoisomerase II Inhibitors: Part 1.

**Figure 3 ijms-24-02532-f003:**
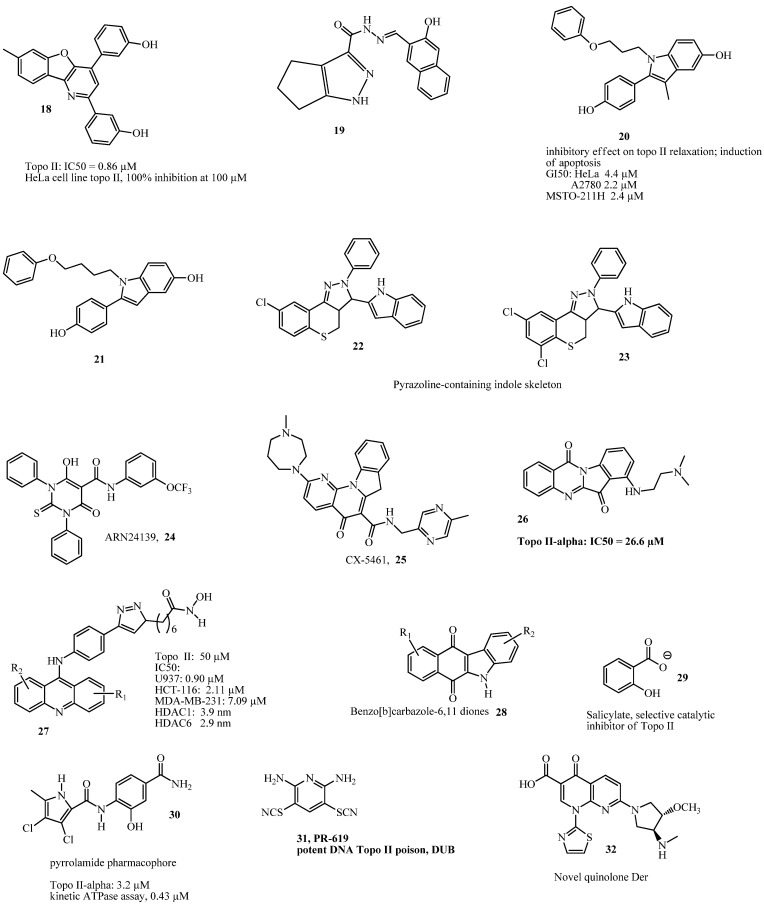
Structures of Topoisomerase Inhibitors: Part 2.

## Data Availability

Not applicable.

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
