# Peer review of "A Mini Review of Novel Topoisomerase II Inhibitors as Future Anticancer Agents"

_ijms, 2023, doi:10.3390/ijms24032532_

Round 1

Reviewer 1 Report

This manuscript “A mini review of novel topoisomerase II Inhibitors as future anticancer agents” is focused on Topoisomerase II that recent publications. Topoisomerase II is an established target for anticancer drugs, that are further subdivided into poisons and catalytic inhibitors. The work review topoisomerase II Inhibitors as future anticancer agents. However, some issues indicated below should be resolved before the paper publication.

1 Summary schematic about topoisomerase II Inhibitors is recommended added in the manuscript.

2 It is recommended to add some review pictures of topoisomerase II Inhibitors anticancer applications, only chemical structural formulas were displayed in this review.

3 There are some spelling and punctuation errors in the manuscript.

Reviewer 2 Report

Line 31-21: Should be under-winding of DNA

Line 49: etoposide binds to the active site and prevents religation of the DNA forming permanent double strand breaks (hence poison) and only in the absence of ATP can it bind to the ATP competitive binding site. The mechanism of doxorubicin and etoposide can be joined together. They both stabilize DNA breaks

Line 72-86. Alot of information in one paragraph. Would suggest providing more detail as you did later in the review if there is more information to be consistent throughout. Later you focus one compound per paragraph.

Figure 2: The formats of the structures are inconsistent. Make sure all in the same format for clarity

Nice review of the compounds on the horizon in the topoisomerase field. 

Reviewer 3 Report

This paper focused on discussing the recent studies finding new topoisomerase II inhibitors other than poisons. 

There are several aspects that need to be considered to fulfill the review. 

1) For the introduction part, studies of TOP2 inhibitors and poisons need to be more thoroughly described. 2) Also, please give a clear description of what is the difference between poisons and inhibitors. 3)Why the review is more focused on the inhibitors other the poisons? What is the advantage of finding inhibitors in comparison to poisons?3) In addition, why there are more studies focusing on finding inhibitors other than poisons? Does that because of the limitation of the assay they addressed? Finding TOP2 inhibitors is much easier than catching TOP2 cleavage complex by poisons. 4) How specific are the TOP2 inhibitors from these studies? 

Listing the reports and studies which are recently been published is fulfill the requirement of a review paper. 

On the other hand, English writing is confusing. In lines 31, and 32, what is under-winning?
